# DP-ViT: A Dual-Path Vision Transformer for Real-Time Sonar Target Detection

Yushan Sun [1], Haotian Zheng [1,*], Guocheng Zhang [1], Jingfei Ren [2], Hao Xu [3] and Chao Xu [4]

1 Science and Technology on Underwater Vehicle Laboratory, Harbin Engineering University, Harbin 150001, China
2 College of Intelligent Systems Science and Engineering, Harbin Engineering University, Harbin 150001, China
3 Marine Design and Research Institute of China, Shanghai 200011, China
4 College of Underwater Acoustic Engineering, Harbin Engineering University, Harbin 150001, China
* Correspondence: zhenghaotian_heu@163.com

**Abstract:** Sonar image is the main way for underwater vehicles to obtain environmental information. The task of target detection in sonar images can distinguish multi-class targets in real time and accurately locate them, providing perception information for the decision-making system of underwater vehicles. However, there are many challenges in sonar image target detection, such as many kinds of sonar, complex and serious noise interference in images, and less datasets. This paper proposes a sonar image target detection method based on Dual Path Vision Transformer Network (DP-VIT) to accurately detect targets in forward-look sonar and side-scan sonar. DP-ViT increases receptive field by adding multi-scale to patch embedding enhances learning ability of model feature extraction by using Dual Path Transformer Block, then introduces Conv-Attention to reduce model training parameters, and finally uses Generalized Focal Loss to solve the problem of imbalance between positive and negative samples. The experimental results show that the performance of this sonar target detection method is superior to other mainstream methods on both forward-look sonar dataset and side-scan sonar dataset, and it can also maintain good performance in the case of adding noise.

**Keywords:** sonar target detection; vision transformer; transformer; convolutional neural network; AUV environment awareness



## 1. Introduction

Autonomous underwater vehicle (AUV) is a kind of equipment that can explore for a long time without human operation. Because of its deep working depth, high efficiency, and long endurance, it has become a research hotspot in various countries. Due to the turbidity and darkness in the underwater environment, the optical image is affected by the color cast, blur, and low line of sight, and the performance is greatly limited. Therefore, the perception system of AUV is mainly completed by sonar. Forward-look sonar can assist AUV to complete oil pipeline inspection, threat detection, mine hunting, and other tasks [1–3]. Side-scan sonar can help AUV find the location of wrecked ships and planes [4,5]. Therefore, the forward-look sonar and side-scan sonar are the most commonly used image sonar carried by AUV. The key technology to accomplish the above tasks is to detect targets in sonar images. Due to the complexity of underwater acoustic channels and the propagation loss of the sound wave itself, sonar images are often characterized by low contrast, serious noise interference, and blurred target contours. Traditional target detection methods are difficult to extract the feature contour of the target accurately. In the AUV submarine pipeline tracking task, if the forward-look sonar image generates false alarms during pipeline detection, the operation efficiency will be reduced. If the target detection accuracy of the side-scan sonar image is low during the USV's search for the wrecked ship, it may lead to missed detection, which may lead to mission failure.

At present, the methods based on deep learning proposed by many researchers can achieve very high target detection efficiency and accuracy in complex scenes. The current mainstream CNN-based target detection networks can be roughly divided into two categories: two-stage and one-stage. Two-stage-based target detection networks such as SPP-Net [6], Faster R-CNN [7], FPN [8], Mask R-CNN [9], R-FCN [10], and so on. The accuracy of target detection is greatly improved, but the detection speed is relatively poor. A one-stage-based network such as SSD [11], CornerNet [12], YOLOV7 [13], etc, May be used. Although the accuracy of this kind of algorithm is lower than that of the two-stage, the real-time detection is very good.

However, in natural language processing (NLP) [14] Transformer, which works very well, was first applied to the computer vision community in 2020, and it has been proved that it still has amazing performance in the CV field. With ViT [15], DETR [16], Swin Transformer V2 [17] Network, the object detection network based on Transformer has been widely used. However, the target detector of the Transformer needs a large number of datasets for training, and it is not suitable for sonar image processing because of its high computational complexity and slow convergence speed. Next-ViT [18] combining CNN with ViT, has achieved very good results in coco dataset. However, there are some problems in sonar images, such as low contrast, loud noise, blurred outline, fewer data, and smaller targets, so the detection accuracy of Next-ViT directly applied in sonar image target detection is not ideal.

Referring to the network structure of Next-ViT, we put forward the Dual Path ViT (DP-VIT) method based on VIT to solve the above problems, which is suitable for target detection in forward-look sonar and side-scan sonar images. In this work, we will propose Dual-Scale Patch Embedding(DSPE), which is subject to Next-ViT [18]. Inspired by the combination of ViT and CNN, the Dual Path transformer Block (DPTB) is proposed, which fully combines the sequence coding ability, global information perception ability of Transformer and generalization ability of CNN. We found that this design can not only keep the translation and rotation invariance of CNN, but also keep the advantages of transformer. The experimental results show that DP-ViT proposed in this paper has better performance than other mainstream target detection methods in side-scan sonar and forward-look sonar target detection tasks, and greatly reduces the influence of low contrast of sonar image and large interference noise on detection tasks. Compared with the single transformer method, it has more obvious advantages in the case of fewer samples. The contributions of this work can be summarized as follows:

Firstly, we propose a new sonar target detection network, called Dual Path ViT, which is suitable for the target detection tasks of side-scan sonar and forward-look sonar. It can reduce the network training parameters on the premise of ensuring the detection accuracy and has a faster convergence speed than the Next-ViT network. It can have better performance on a sonar dataset with fewer samples.

Secondly, Dual-Scale Patching Embedding is adopted instead of the original Patching Embedding. tokens of different scales will be sent to Transformer Encoder in parallel, and Transformer Encoder with different patch sizes will execute global self-attention. The proposed DSPE can effectively enlarge the receptive field and reduce the influence of noise on the accuracy of the target detection task.

Thirdly, refer to Next-ViT [18], according to DSPE, the Dual Path Transformer Block is designed, which connects the transformative local features to the transformer global features and connects the local features with the global features. The proposed DPTB can better obtain the global and local feature information of images and can obtain higher detection accuracy in complex sonar images.

Fourthly, the sonar target image dataset of forward-look sonar and side-scan sonar are constructed respectively, and the advanced performance of DP-ViT in sonar image target detection is verified on this dataset.

The rest of the paper is organized as follows: in Section 2, which briefly introduces the frontier work done by other researchers in this field. In Section 3, the principles of DSPE

and DPTB and other structures of DP-ViT are introduced. In Section 4, the training results of DP-ViT are given, compared and discussed with other methods. Finally, the conclusion and pointing-out of future directions are given in Section 5.

## 2. Related Works

**Sonar Target Detection**. Sonar image has serious noise interference and complex background, which will affect the target detection algorithm to some extent. Many researchers have proposed many different sonar image target detection algorithms, among which it is difficult to effectively and accurately detect targets in sonar images based on traditional machine learning frameworks and traditional image processing methods. For example, in [19,20], traditional methods are used to detect underwater targets, in terms of detection accuracy and real-time, traditional target detection methods are inferior to depth learning methods [21], and the detection speed can not meet the real-time requirements.

Other researchers have also proposed and improved many sonar target detection methods based on deep learning. Kim and Yu [22], applying the deep learning object detection model YOLO to the detection and tracking of sonar targets. Kong et al. [23] improved on YOLOv3 by proposing YOLOv3-DPFIN (Dual-Path Feature Fusion Neural Network) for real-time object detection of sonar images. Fan [24] proposed Detection and segmentation of underwater objects from forward-look sonar modified Mask RCNN. However, in the above methods, they only studied the detection of side-scan sonar or foresight sonar, and real-time and accuracy were not well balanced.

**Convolutional Networks**. In the past few years, Convolutional Networks (CNNs) have played a leading role in computer vision tasks, including image classification, object detection, semantic segmentation, and image enhancement. MobileNetV1 [25] first proposed depthwise separable filters to build lightweight Convolutional Neural Networks. ShuffleNet [26] proposes a pointwise group convolution and channel shuffle that can achieve a significant reduction in the amount of computation at the same precision and further reduce the cost of computation. Although ConvNeXt [27] did not have much innovation in the overall network framework and construction ideas, he introduced part of the transformer idea into the network of CNNs, and still ensured the simplicity and efficiency of the CNN structure while improving accuracy. The Sparse R-CNN [28] network structure is extremely simple, eliminating the need for dense anchors, RPNs, and complex post-processing and NMS, with fast convergence and efficient detection efficiency.

**The Vision Transformer.** Transformer was first applied to natural language processing (NLP). ViT [15] is the first method of transformer to be applied in the field of computer vision, and it proves that transformers have more powerful global information perception capabilities than CNNs. DeiT [29] introduced a distillation token to implement a teacher-student strategy that does not require external data. Deformable DETR [30] introduces the Deformable Attention Module compared to DETR, which solves the problems of long training cycles and slow convergence in DETR. Li et al. [31] first applied ViT as a backbone to the field of object detection with good results. Next-ViT [18] strikes a balance between real-time and accuracy.

## 3. Method

In the process of sonar image target detection, whether it is side-scan sonar or forward-look sonar, it is necessary to extract the features of the target area in the sonar image to determine the position and category of the target. Although the above methods based on CNN and Transformer have a good performance in the task of target detection. However, there are few sonar image datasets, and the strong noise of sonar echo results in low image SNR, the above network has some problems, such as high computational complexity, poor generalization, low detection accuracy, slow convergence speed, and difficulty insufficient training.

Inspired by the multi-scale task, we propose a Dual-Scale Patch Embedding to replace the Patch Embedding module in Next-ViT. Different from the existing multi-scale tasks,

DSPE can have better feature extraction and feature fusion capabilities at the same sequence length and can pass the two-scale Receptive Field Acquisitions Global Context Information. This design can better improve the generalization ability of the model and is more conducive to the application of small target detection in sonar images with low signal-to-noise ratio and few samples.

Compared with the coco dataset, both the forward-look sonar dataset and the side-scan sonar dataset are simpler in terms of the number and size of target objects. Therefore, we think that the design of NCB and NTB in Next-ViT is complicated in sonar image detection. In order to save the cost of computing resources, this paper combines NCB and NTB modules into Dual Path Transformer Block (DPTB) and improves it according to the proposed DSPE module. This design can reduce the computational complexity of the model, improve the convergence speed, and is more suitable for sonar image target detection.

In this section, firstly, the DP-ViT proposed in this paper is introduced. Then, some special designs in DP-ViT will be discussed, including dual-scale patching embedding (DSPE), Dual Path Transformer Block (DPTB), and Loss Function.

### 3.1. Overview

We present the DP-ViT as illustrated in Figure 1. Same as most other networks, DP-ViT follows the hierarchical pyramid architecture equipped with a Dual-Scale Patching Embedding layer and a Dual Path Transformer Block in each stage. Similar to the overall structure design of Next-ViT, the spatial resolution will be progressively reduced by $32\times$ while the channel dimension will be expanded across different stages. In this chapter, we first discuss deeper designing the core blocks for information interaction and elaborate on the proposed DSPE and DPTB. At the same time, Transformer (global information) and CNN (local information) also perform information fusion in DPTB. Finally, due to the linear complexity of the Dual-Path structure, it requires more resources for computation. We use factorized self-attention as done in CoaT [32] and generalized focal loss as done in GFL [33].

### 3.2. Dual-Scale Patching Embedding (DSPE)

There are four stages in our network, and each stage includes DSPE and DPTB. Because of the Dual-scale structure, Conv-stem was designed before the first stage, and the dimension was reduced by two downsamplings. This operation can not only reduce the amount of calculation and the length of the sequence but also will not cause the sequence length to be too short, as in ViT, which is not conducive to feature extraction.

We designed a Dual-Scale Patch Embedding (DSPE) layer, which independently inputs dual-scale tokens into DPTB by embedding patches at the same time. We introduce overlapping matches, similar to TransFG [34]. Given the input image size is $H \times W \times 3$, $stage_i$. The input of is $X_i \in \mathbb{R}^{H_{i-1} \times W_{i-1} \times C_{i-1}}$, then the output token $map\ F_{k \times k}(X_i) \in \mathbb{R}^{H \times W \times C}$ Has height and width are as follows:

$$H_i = \frac{H_{i-1} - k + 2p}{s} + 1 \tag{1}$$

$$W_i = \frac{W_{i-1} - k + 2p}{s} + 1 \tag{2}$$

In which kernel size $k$, stride $s$, and padding $p$. According to the above formula, the sequence length of tokens can be changed by adjusting stride and padding. That is to say, the same size output features can be generated by different patch sizes. So we use $3\times3$ and $5\times5$ Double kernel sizes, to generate the convolutional patch embedding layers in parallel, such as Figure 2 shown. By embedding a patch of the same size at the same time, tokens of different scales are input into DPTB through independent paths, so as to achieve rough and fine feature representation on the same level of features. Using stacking consecutive convolution operations, you can use fewer parameters to obtain the larger receptive field, we use two consecutive $3\times3$ convolutions with the same padding, stride, and channel

size instead of a 5×5 Convolutions S. due to the dual-path architecture adopted, DP-ViT will have more embedding layers. We adopt 3×3 depthwise separable convolutions [35] to reduce model parameters and reduce computation. Finally, the double sizes token embedding is sent to the DPTB separately.

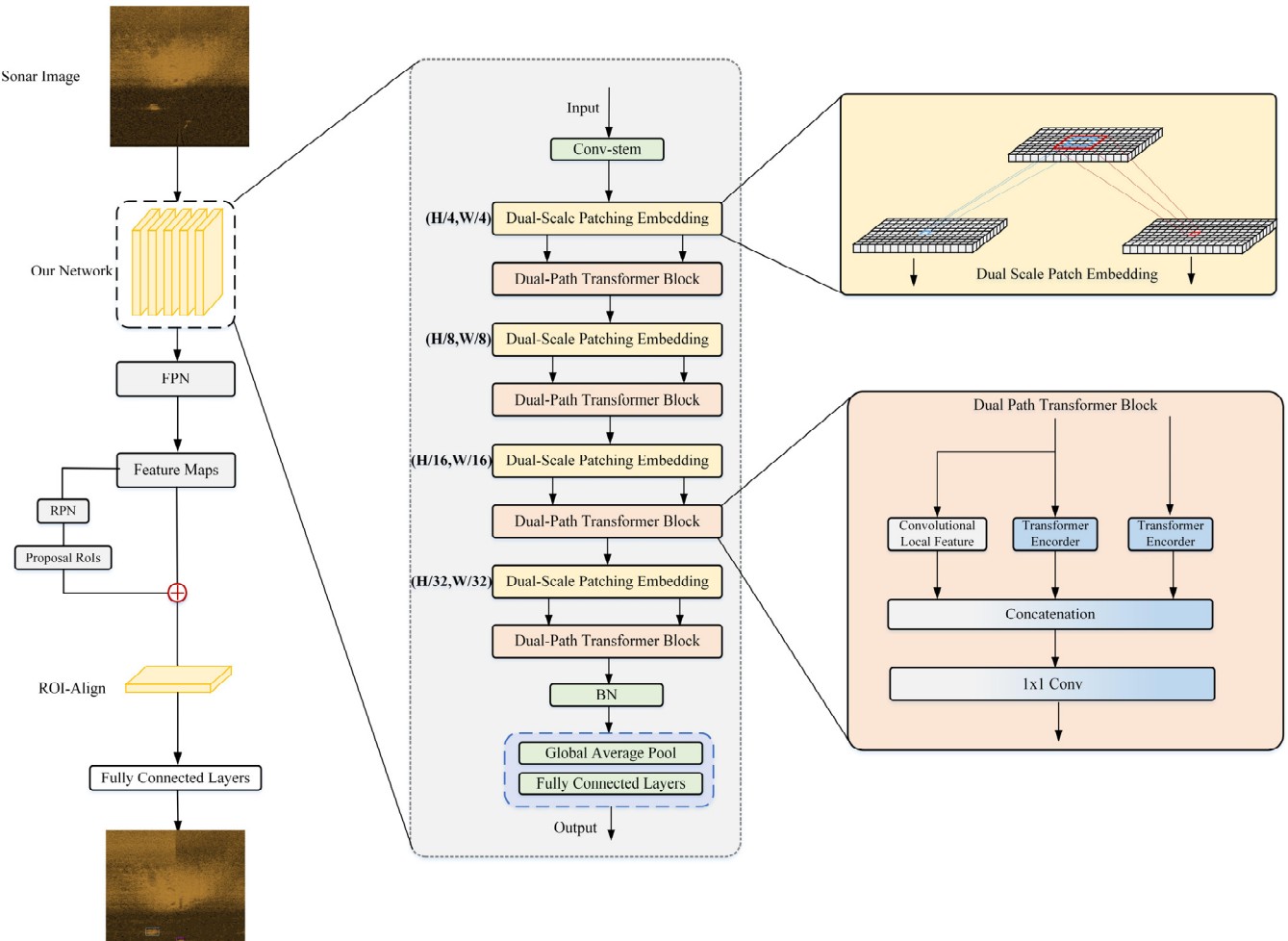

**Figure 1.** The middle column of the proposed DP-ViT framework includes Dual Scale Patching Embedding and Dual Path Transformer Block. On the right is the detailed visualization of dual scale and transformer block in patch embedding.

### 3.3. Dual Path Transformer Block (DPTB)

In some classical structural designs of CNN and Transformer block, such as Resnet [36], eliminates the problem of "degradation", that is, the difficulty of training neural networks with excessive depth, and guides scholars to develop neural networks to "depth". However, Neck Block is still low in effectiveness compared to Transformer Block. ConvNeXt [27] improved Neck Block and improved network performance, the low-efficiency modules such as GELU and LayerNorm are used, which makes it impossible to complete real-time target detection. Transformer has made great achievements in all major visual tasks, but its complicated attention mechanism and sequence expansion seriously affect the reasoning speed. Next Evolution Block (NCB) and Next Transformer Block (NTB) are introduced in Next-ViT [18]. Next-ViT proves that this design has both excellent performance and good real-time performance. Because the number of classes in the sonar image target detection task is small and the background noise of the image does not change much, we believe that this design is redundant in the sonar image target detection task and does not perform well in the detection task of smaller targets. In order to solve the above problems, we

introduced the Dual Path Transformer Block (DPTB) and fused local information with global information, which further improved the generalization of the model, reduced the required computing resources, and improved the reasoning speed.

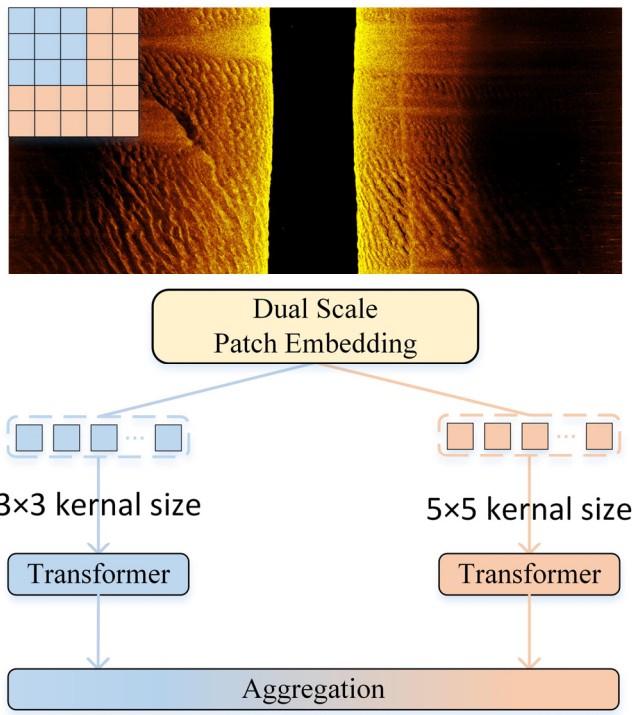

**Figure 2.** The dual path structure shows that each embedded patch will enter an independent transformer block.

### 3.3.1. Transformer Encoder

In the field of target detection, Transformer block has high precision, and its ability to capture global information is unique. For example, it has a strong ability to capture low-frequency signals of global shapes and global structures. However, due to its complex attention mechanisms, its real-time performance and reasoning speed have been seriously affected. In order to overcome the above shortcomings, we developed a Tranformer Encoder to capture the shape and structure information in sonar images, and further enhanced the modeling capability. Although depthwise separable convolutions are used in the DSPE structure, it will still increase the training parameters and calculation amount of the model. In order to solve the above problems and reduce the impact of DSPE on model training, the method shown in Figure 3 is proposed. Firstly, the efficient Factorized MHSA is used to replace the original Self Attention. CoaT designed the Factorized Attention Block in a convolution-like way, introduced co-scale and designed Conv-Attention, and embedded the relative position in the factored attention. Through multi-scale modeling, the learning ability of the model was improved and the parameters were greatly reduced [32]. The Conv-Attention mechanism can keep the integrity of Transformer Encoder at all scales and can add multi-scale information and context modeling functions. Factorized self-attention proposed in CoaT:

$$FactorAtt(Q, K, V) = \frac{Q}{\sqrt{C}}(softmax(K)^T V) \tag{3}$$

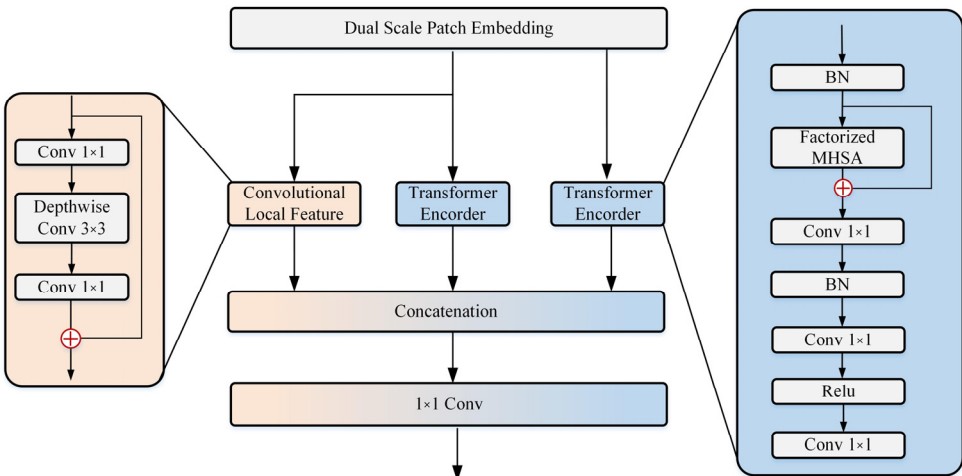

**Figure 3.** Overview of Dual Path Transformer Block.

Among $Q, K, V \in R^{N \times C}$ is the queries, keys, and values of the linear projection. N and C represent the number and embedded dimension of tokens, respectively. If there is no convolution position coding, the Transformer is only composed of the self-attention module, which will result in the model being unable to obtain the difference of local features, and this feature cannot be applied to the sonar image target detection task with large background noise. In order to be able to better integrate CoaT into DP-ViT, according to the class token and image token in ViT [15]. We use 2-D depthwiseconv convolution, which is only used to calculate image token (i.e., $Q^{img}, V^{img} \in \mathbb{R}^{H \times W \times C}$ from $Q, V$ respectively):

$$ConvAtt = FactorAtt(Q, K, V) + concat(Q^{img} \circ DepthwiseConv2D(P, V^{img}), 0) \quad (4)$$

Among them, $\circ$ is Hadamard product. At the same time, replacing LN and GELU in a traditional transformer with BN and ReLU not only speeds up the calculation speed but also improves the model performance.

### 3.3.2. Feature Interaction

A related study [37] shows that transformer block will worsen high-frequency information such as local textures information to some extent, ignoring the structural information and local relationship in each patch. However, this information is indispensable in sonar image target detection. To avoid the loss of local features caused by the above problems, the Convective Local Feature is introduced into DPTB. The local connectivity in translation invariance and rotation invariance of CNN is used to compensate for the influence of Transformer on the model. To represent local features $X_{L_{i-1}} \in \mathbb{R}^{H_{L_{i-1}} \times W_{L_{i-1}} \times C_{L_{i-1}}}$, we construct a Depth wise residual bottleneck block, including 1×1 convolution, 3×3 depth convolution, 1×1 convolution, and residual connection. DPTB connects CNN and Transformer in a complementary way. Therefore, we introduced a global feature and local feature fusion module, which fused the global feature and local feature together by aggregation and series connection.

### 3.4. Loss Function

There are some problems in the sonar image dataset, such as unbalanced positive and negative samples, a small number of samples, and more image noise. So that a part of the scores in the training process is low, and the quality prediction which is actually a positive sample but is judged as a "negative sample" cannot be defined in the training process. As a result, it is likely that a real negative sample (such as noise in the image) will be predicted as a high-quality score, which will lead to the problem that the predicted score of the real target that should be identified and detected in the sonar image is lower than that of the

negative sample such as noise. Therefore, we introduce Generalized Focal Loss (GFL) [33] to solve the above problems. The formula of GFL is:

$$GFL = -\left|y - (y_l P_{y_l} + y_r P_{y_r})\right|^{\beta} \left((y_r - y) \log(P_{y_l}) + (y - y_l) \log(P_{y_r})\right) \quad (5)$$

Among them, $y \in \{1, 0\}$ specifies the ground truth classes, $p \in [0, 1]$ is the estimated probability, $\gamma$ is a parameter.

## 4. Experiment

### 4.1. Model Training

In order to verify the effectiveness and superiority of DP-ViT, this paper conducts experimental comparative analysis and verification on the datasets of forward-look sonar and side-scan sonar respectively. The original images of forward-look sonar are generally presented in the form of acoustic reflection, which are all grayscale images [38,39]. The forward-look sonar dataset used in this paper was collected using AUV-R in Changling Lake, Harbin, China, and the sonar equipment used is M1200D forward-look sonar. Figure 4 shows the AUV-R. The dataset includes eight kinds of objects (cube, sphere, cylinder, human body, tire, annular cage, and iron barrel). The images in the dataset are the original echo images of the forward-look sonar after gain and coloring, but without polar coordinate transformation, which not only helps to keep more details of sound waves, but also facilitates the annotation and manual interpretation of the dataset. This dataset consists of 1650 training images and 350 test images, and the labeling format is VOC.

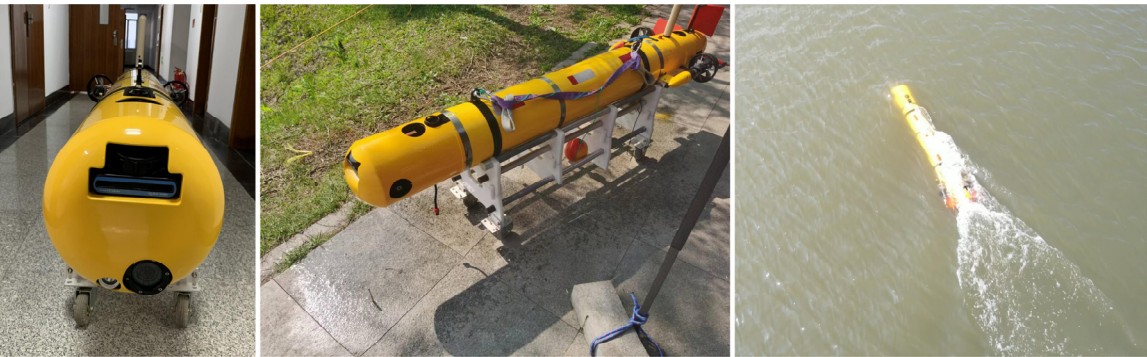

**Figure 4.** AUV-R physical picture and experimental picture.

The dataset of side-scan sonar images is publicly available on the Internet, includes three kinds of graph-free (human, ship, aircraft), which are composed of 650 training images and 250 test images, and the annotation format is VOC. This dataset has serious seabed noise and reverberation interference. Aiming at the problems of complex background and loud noise of sonar images, this paper adds Gaussian noise, salt and pepper noise, and Rayleigh noise to the test sets of two datasets, respectively, to verify that DP-ViT algorithm still has good performance in the case of loud noise, which proves that the model has good generalization.

In this paper, a small batch gradient descent algorithm is used to optimize the parameters. The small batch gradient descent method is a compromise between the batch gradient descent method and the random gradient descent method. The gradient is used to determine the direction of the parameter update. In the process of each iteration, "batch size" samples are used to update the parameters, and the parameters of the objective function are updated repeatedly, so that the objective function gradually approaches the minimum value. Small batch gradient descent can be accelerated by matrix and vector calculation, and the variance of parameter update can be reduced to obtain more stable convergence. When the batch size is selected reasonably, the small batch gradient descent method can

improve the memory utilization, reduce the iteration times of each epoch, further accelerate the convergence speed and reduce the training shock. Specific process reference Figure 5.

```
Train a neural network with mini-batch stochastic gradient descent
    initialize(net)
    for epoch = 1,...,K do
        for batch = 1,...,#images/b do
            images←uniformly random sample b images
            X,y←preprocess(images)
            z←forward(net,X)
            L←loss(z,y)
            grad←backward(L)
            update(net,grad)
        end for
    end for
```

**Figure 5.** Overview of the pseudo-code flow chart of the mini-batch descent gradient.

Albumentations are used in sonar dataset training [40]. The input image trained by the model is randomly rotated, flipped, translated, scaled, and other image transformation operations to prevent the occurrence of over-fitting.

In order to test the target detection performance of the network and compare it with other networks, we use Precision, Recall, Mean Average Precision (*mAP*), GFLOPS and FPS to quantitatively evaluate the target detection performance of DP-ViT and other networks on the sub-test set. GFLOPs is Giga Floating-point Operations, which is used to quantitatively measure the complexity of the model. FPS is used to evaluate the speed of the target detection model, that is, the number of images that can be processed per second or the time required for one image. Precision and recall are calculated as follows:

$$precision = \frac{TP}{TP + FP} \tag{6}$$

$$recall = \frac{TP}{TP + FN} \tag{7}$$

where *TP* is the real positive samples predicted by the model as positive samples. *FP* is a negative sample predicted to be positive by the model. *FN* is a positive sample with a negative model prediction. The *mAP* is equal to taking the area under the precision-recall curve. The paper uses the *mAP* calculation standard in VOC2012. The *AP* and *mAP* calculation formula is as follows:

$$AP = \sum_{r=0}^{1} (r_{n+1} - r_n) P_{\text{interp}}(r_{n+1}) \tag{8}$$

$$mAP = \frac{\sum_{n=1}^{N} AP_n}{N} \tag{9}$$

All the models in this paper are built on the mature mmdetection framework using Python. The operating system of the platform is Ubuntu18.04, computer memory is 32 GB, Nvidia RTX3090 graphics card is used as hardware, and Intel Core i9-10900 CPU is equipped.

### 4.2. Experiments on Forward-Look Sonar

In order to verify the performance of DP-ViT network, we compare DP-ViT with other detection methods on the forward-look sonar image dataset, including some general target detection methods such as Faster R-CNN [7], YOLOX [41], Sparse R-CNN [28],

Next-ViT [18], and the sonar target detection model YOLOv3-DPFIN [23]. The test results are shown in Figure 6. The confidence levels of each method are described in Table 1. The training results of each detection method in the forward-look sonar dataset are as shown in Table 2.

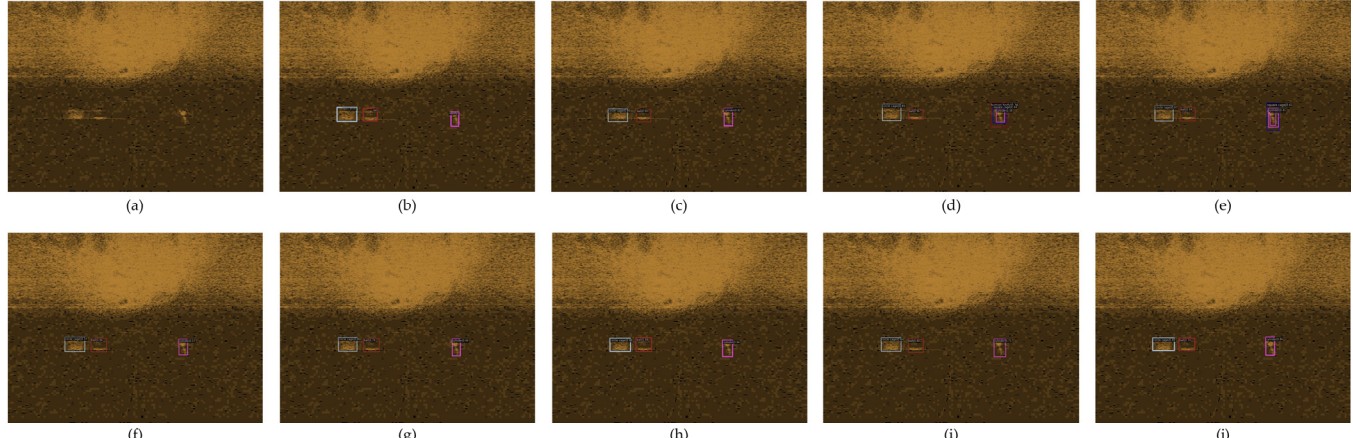

**Figure 6.** Comparison of DP-ViT and other methods in forward-look sonar dataset (**a**) real sonar image (**b**) true sonar image target (**c**) our method: DP-ViT (**d**) Faster R-CNN (Resnet50) (**e**) Faster R-CNN (Resnet101) (**f**) Sparse R-CNN (resnet50) (**g**) Sparse RCNN (resnet101) (**h**) Next ViT (**i**) YOLOX-s (**j**) YOLOv3 DPFIN.

**Table 1.** Figure 6 Schematic test results.

| Fig | Method | Actual Sonar Targets | True Sonar Targets Detected | Confidence (AP) |
|-----|--------|:--------------------:|:--------------------------:|:---------------:|
| (c) | DP-ViT | 3 | 3 | 0.90; 0.94; 0.82 |
| (d) | Faster R-CNN (Resnet50) | 3 | 3 | 0.85; 0.96; 0.38 |
| (e) | Faster R-CNN (Resnet101) | 3 | 3 | 0.97; 0.94; 0.61 |
| (f) | Sparse R-CNN (Resnet50) | 3 | 3 | 0.77; 0.86; 0.57 |
| (g) | Sparse R-CNN (Resnet101) | 3 | 3 | 0.80; 0.74; 0.84 |
| (h) | Next-ViT | 3 | 3 | 0.88; 0.88; 0.94 |
| (i) | YOLOX-s | 3 | 3 | 0.64; 0.80; 0.71 |
| (j) | YOLOv3 DPFIN | 3 | 3 | 0.87; 0.74; 0.84 |

It can be seen from Figure 6 and Table 1 that DP ViT has the highest confidence level of 0.90, 0.94 and 0.82 in the forward look sonar image target detection task. Table 2 shows that DP-ViT has a relatively small model size of 59.82 M (Param), a small parameter quantity of 145.23 G (FLOPs), a higher detection accuracy of 89.2% (*mAP*) and a better real-time performance of 43.9 (FPS).As the *mAP* and Confidence of YOLOX-s in sonar image target detection task are poor, the comparison result of the loss curve is not included in YOLOX, and the comparison chart of LOSS is shown in Figure 7.

### 4.3. Experiments on Side-Scan Sonar

In order to further verify the performance of the proposed DP-ViT in target detection on other types of sonar images, we verify the DP-ViT on the side scan sonar dataset. The test results are shown in Figure 8. The confidence of each method is described in Table 3. The training results of each detection method in the forward look sonar dataset are shown in Table 4.

**Table 2.** Detection Build on Forward-Look Sonar, training and testing on the forward-look sonar dataset and evaluating each algorithm on RTX3090 using CUDA11.3 and CUDNN8.2.1.

| Method | mAP | APs | Stop (M) | FLOPs (G) | FPS | Precision | Recall |
|---|---|---|---|---|---|---|---|
| DP-ViT | 89.2 | 87.8 | 59.82 | 145.23 | 43.9 | 92.4 | 91.1 |
| Faster R-CNN (Resnet50) | 82.4 | 81.3 | 60.15 | 201.95 | 37.1 | 86.5 | 83.5 |
| Faster R-CNN (Resnet101) | 86.1 | 85.5 | 81.1 | 282.77 | 26.5 | 85.9 | 89.0 |
| Sparse R-CNN (Resnet50) | 79.7 | 77.4 | 105.95 | 169.9 | 40.8 | 77.8 | 77.4 |
| Sparse R-CNN (Resnet101) | 88.6 | 86.5 | 124.95 | 225.97 | 33.6 | 87.9 | 86.3 |
| Next-ViT | 81.2 | 83.1 | 61.28 | 159.05 | 41.5 | 83.7 | 84.4 |
| YOLOX-s | 72.2 | 71.3 | 25.28 | 91.9 | 68.6 | 73.9 | 72.5 |
| YOLOv3 DPFIN | 84.8 | 79.4 | 43.4 | 105.5 | 56.2 | 83.9 | 80.5 |

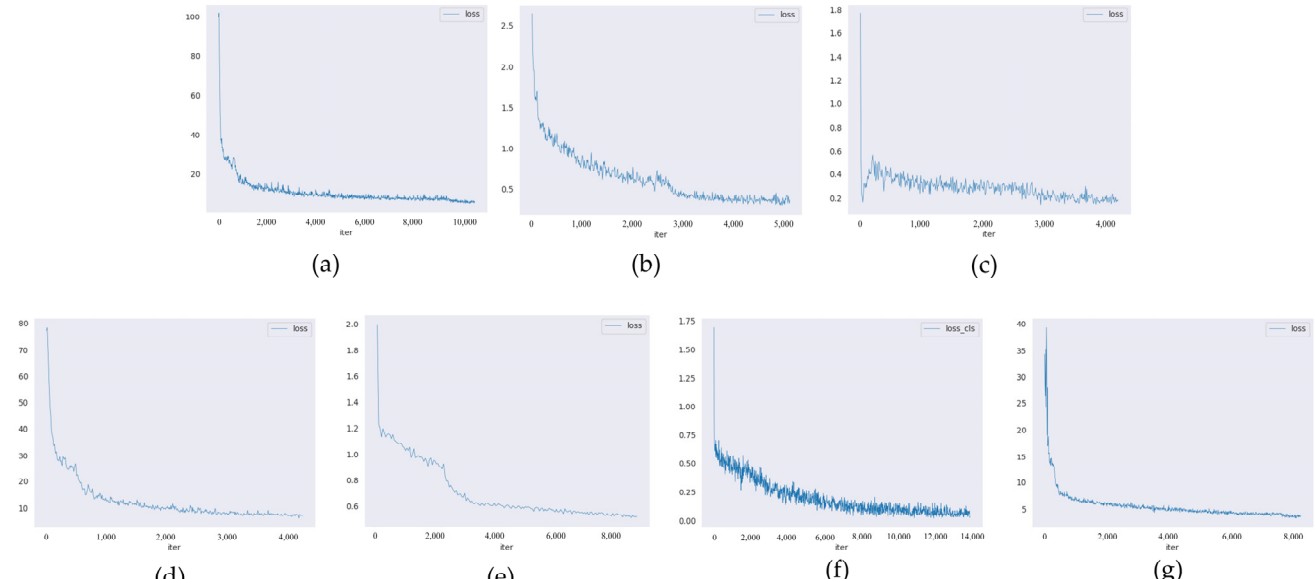

**Figure 7.** Loss comparison chart of each method.(**a**) Our method:DP-ViT (**b**) Faster R-CNN(Resnet 50) (**c**) Faster R-CNN(Resnet 101) (**d**) Sparse R-CNN(resnet 50) (**e**) Sparse R-CNN(resnet 101) (**f**) Next-ViT (**g**) YOLOv3 DPFIN.

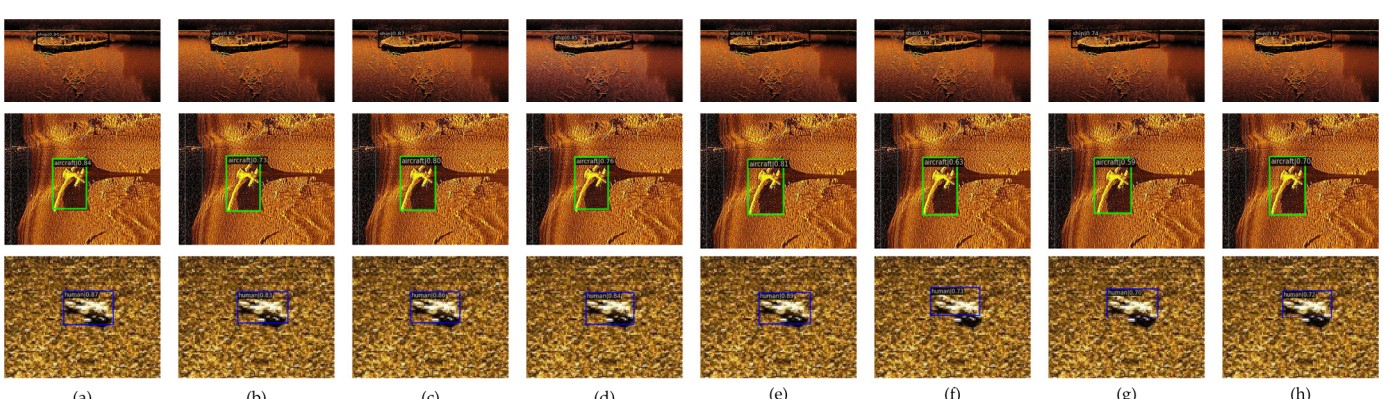

**Figure 8.** Comparison of DP-ViT and other methods in side-scan sonar dataset (**a**) our method:DP-ViT (**b**) Faster R-CNN (Resnet50) (**c**) Faster RCNN (Resnet101) (**d**) Sparse R-CNN (resnet50) (**e**) Sparse R-CNN (resnet101) (**f**) Next ViT (**g**) YOLOX-s (**h**) YOLOv3 DPFIN.

**Table 3.** Figure 7 Schematic test results.

| Method | DP-ViT | Faster R-CNN (Resnet50) | Faster R-CNN (Resnet101) | Sparse R-CNN (Resnet50) |
|---|---|---|---|---|
| Confidence (AP) | human: 0.87 Aircraft: 0.84 Ship: 0.90 | human: 0.83 Aircraft: 0.73 Ship: 0.82 | human: 0.86 Aircraft: 0.80 Ship: 0.97 | human: 0.84 Aircraft: 0.76 Ship: 0.85 |
| Method | Sparse R-CNN (Resnet101) | Next-ViT | YOLOX-s | YOLOv3 DPFIN |
| Confidence (AP) | human: 0.91 Aircraft: 0.81 Ship: 0.89 | human: 0.73 Aircraft: 0.63 Ship: 0.79 | human: 0.70 Aircraft: 0.59 Ship: 0.74 | human: 0.72 Aircraft: 0.70 Ship: 0.82 |

**Table 4.** Detection Build on Side-Scan Sonar, training and testing on Side-Scan Sonar dataset and evaluating each algorithm on RTX3090 using CUDA11.3 and CUDNN8.2.1.

| Method | mAP | APs | Stop (M) | FLOPs (G) | FPS | Precision | Recall |
|---|---|---|---|---|---|---|---|
| DP-ViT | 85.6 | 85.4 | 59.82 | 145.23 | 43.4 | 85.4 | 85.7 |
| Faster R-CNN (Resnet50) | 79.1 | 79.5 | 60.15 | 201.95 | 37.6 | 78.6 | 79.0 |
| Faster R-CNN (Resnet101) | 83.3 | 83.5 | 81.1 | 282.77 | 27.1 | 82.3 | 83.2 |
| Sparse R-CNN (Resnet50) | 80.4 | 81.1 | 105.95 | 169.9 | 42.1 | 79.7 | 79.6 |
| Sparse R-CNN (Resnet101) | 84.2 | 85.5 | 124.95 | 225.97 | 34.6 | 82.3 | 83.4 |
| Next-ViT | 76.2 | 77.5 | 61.28 | 159.05 | 41.7 | 77.1 | 77.5 |
| YOLOX-s | 72.4 | 73.1 | 25.28 | 91.9 | 67.9 | 73.2 | 72.1 |
| YOLOv3 DPFIN | 79.4 | 78.7 | 43.4 | 105.5 | 56.5 | 78.3 | 79.2 |

In the side scan sonar dataset test, it can be seen from Figure 8 and Table 3 that DP-ViT has the highest confidence level of 0.87, 0.84 and 0.90. Table 4 shows that DP ViT has the highest detection accuracy of 85.6% (*mAP*). As the *mAP* and Confidence of YOLOX-s in the target detection task of side scan sonar image are poor, the comparison result of the loss curve is not included in YOLOX-S, and the comparison chart of LOSS is shown in Figure 9 below.

### 4.4. Experiments with Different Noise

Because sonar will attenuate, reverberate and scatter when propagating in water, there are usually a lot of different types of noises in sonar images. According to different noise sources, they can be divided into three categories: (1) Because the environmental noise caused by the movement of marine medium, the change of water body characteristics and the sound emitted by marine organisms will affect the propagation of sound waves in the water, thus affecting the accuracy of sonar images, this paper uses Gaussian noise simulation. (2) Because there are a large number of suspended solids and scatterers in the marine environment, which interfere with the real target echo, this paper adopts salt and pepper noise simulation. (3) Reverberation noise is the most important interference signal of sonar image, and even covers the real target when it is serious. According to Middleton [42], based on the proposed seabed reverberation model, it can be considered that the phase of reverberation obeys uniform distribution, and the amplitude characteristics conform to the Rayleigh distribution. Rayleigh noise can be realized according to the following formula.

$$X = \eta + \sqrt{-\mu \ln[1 - U(0,1)]} \tag{10}$$

where $\eta$ and $\mu$ represent noise intensity and spot size, respectively. In order to further verify the anti-noise ability of the algorithm in this paper, Rayleigh noise, Gaussian noise, and salt and pepper noise transform are respectively carried out on the image. The sonar images with different noises are shown in Figure 10, where the first three columns are side scan sonar images and the last three columns are forward look sonar images. Under the condition of increasing noise, the network model trained by a training set without noise is adopted to test the sonar image of DP-ViT with noise, and the test results are shown in Table 5.

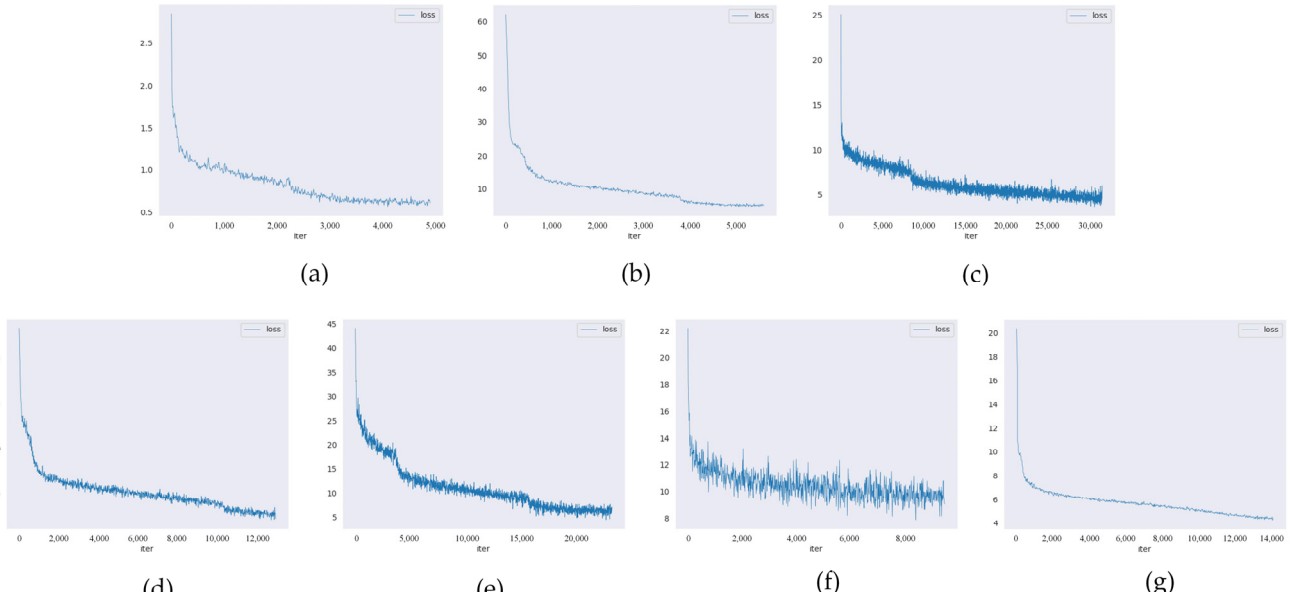

**Figure 9.** Loss comparison chart of each method. (**a**) Our method:DP-ViT (**b**) Faster R-CNN (Resnet 50) (**c**) Faster R-CNN (Resnet 101) (**d**) Sparse R-CNN (Resnet 50) (**e**) Sparse R-CNN (Resnet 101) (**f**) Next-ViT (**g**) YOLOv3-DPFIN.

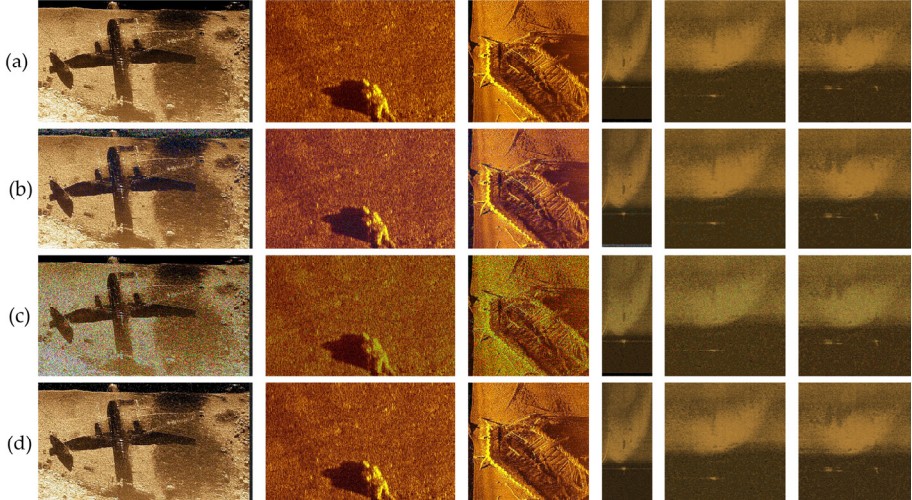

**Figure 10.** Schematic diagram of adding noise to side-scanning sonar image and front looking sonar image. (**a**) Real image (**b**) Gaussian noise (**c**) Rayleigh noise (**d**) Salt and pepper noise.

**Table 5.** Test results of DP-ViT, YOLOv3 DPFIN, and Sparse R-CNN (Resnet101) in different noise environments (FLS Dataset: Forward-Look Sonar Dataset, SSS Dataset: Side-Scan Sonar Dataset).

| Method | | | mAP | APs | Precision | Recall |
|---|---|---|---|---|---|---|
| DP-ViT | FLS Dataset | Gaussian noise | 87.1 | 86.9 | 88.2 | 87.9 |
| | | Rayleigh noise | 80.3 | 79.5 | 80.6 | 80.4 |
| | | Salt and pepper noise | 88.2 | 87.3 | 89.4 | 89.1 |
| | SSS Dataset | Gaussian noise | 81.3 | 80.3 | 80.2 | 80.7 |
| | | Rayleigh noise | 71.5 | 70.8 | 70.7 | 71.3 |
| | | Salt and pepper noise | 84.1 | 85.0 | 84.7 | 84.9 |
| YOLOv3 DPFIN | FLS Dataset | Gaussian noise | 82.5 | 81.7 | 80.1 | 79.8 |
| | | Rayleigh noise | 78.3 | 79.1 | 76.3 | 76.0 |
| | | Salt and pepper noise | 81.6 | 82.5 | 79.4 | 80.1 |
| | SSS Dataset | Gaussian noise | 74.2 | 73.5 | 74.6 | 73.9 |
| | | Rayleigh noise | 72.4 | 72.1 | 71.7 | 71.7 |
| | | Salt and pepper noise | 73.9 | 73.5 | 73.4 | 73.0 |
| Sparse-RCNN (Resnet101) | FLS Dataset | Gaussian noise | 87.7 | 86.8 | 85.3 | 86.9 |
| | | Rayleigh noise | 84.2 | 83.0 | 81.7 | 82.5 |
| | | Salt and pepper noise | 87.1 | 88.4 | 87.5 | 88.3 |
| | SSS Dataset | Gaussian noise | 82.3 | 82.9 | 81.5 | 81.8 |
| | | Rayleigh noise | 75.3 | 75.6 | 73.7 | 73.9 |
| | | Salt and pepper noise | 81.5 | 80.7 | 79.5 | 80.2 |

From the data in Table 5, it can be found that noise interference has a greater impact on the detection accuracy of side-scan sonar images and has less impact on the detection accuracy of forward-look sonar images. From the experimental results, Sparse R-CNN is less affected by noise interference. Our analysis suggests that Proposal Boxes and Proposal Features play a significant role in anti-noise interference, but their complex network structure cannot realize real-time target detection task on AUV. DP-ViT still has high accuracy under strong noise interference condition, and has good regression accuracy and robustness in a low-SNR environment.

*4.5. Experiments with Fewer Sonar Samples Dataset*

In order to verify the performance of DP-ViT in the few-sample sonar data set, a small number of images were randomly selected as the training set in the forward-looking sonar data set and the side-scan sonar data set to test the detection accuracy of the DP-ViT network in the case of few samples. The constructed forward-looking sonar dataset includes 160 images as training set and 40 images as test set. The side-scan sonar dataset includes 80 images as training set and 20 images as test set. The test results are shown in Figure 11 and Table 6 below.

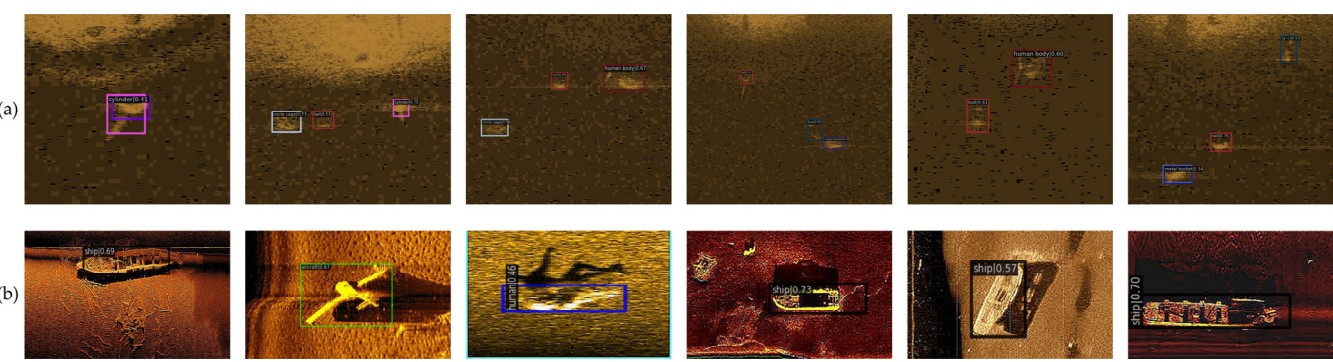

**Figure 11.** Test Results of DP-ViT on Sonar Images with Few Samples. (**a**) Forward-Look Sonar Dataset (**b**) Side-Scan Sonar Dataset.

**Table 6.** Test results of DP-ViT on Sonar Images with Few Samples (FLS Dataset: Forward-Look Sonar Dataset, SSS Dataset: Side-Scan Sonar Dataset).

| Dataset | mAP | APs | Precision | Recall |
|---|---|---|---|---|
| FLS Dataset | 70.9 | 71.3 | 71.7 | 73.8 |
| SSS Dataset | 68.5 | 67.8 | 69.1 | 69.8 |

From the experimental results, DP-ViT has a good performance in the case of small sample training. Although the confidence has decreased, it can still complete the target detection task.

*4.6. Ablation Study*

In order to verify the influence of the four transformation methods on the training model, this paper adopts the control variable method, which changes the Dual Scale Patch Embedding into Patch Embedding, removes the Convolution Local Feature in the Dual Path Transformer Block, and removes the Generalized Focal Loss. MAP and APs are obtained by verification in the forward look sonar dataset, and the experimental results are shown in Table 7. In addition, it can be found that when Dual Scale Patch Embedding is removed and replaced with ordinary Patch Embedding, it will have a greater impact on target detection accuracy.

**Table 7.** Effectiveness of Various Designs.

| Method | DP-ViT | | | |
|---|---|---|---|---|
| Dual Scale Patch Embedding? | √ | | √ | √ |
| Convolution Local Feature? | √ | √ | | √ |
| Generalized Focal Loss? | √ | √ | √ | |
| mAP | 89.2 | 82.4 | 87.9 | 85.3 |
| APs | 87.8 | 79.1 | 85.6 | 84.0 |

*4.7. Qualitative Assessment*

The DP-ViT method proposed in this paper has good performance in target detection tasks of both forward-look sonar and side-scan sonar, and its performance in mAP, APs, and other indicators is superior to other mainstream target detection networks. On the convergence speed of the training model, because the network based on Transformer has some disadvantages, such as difficult training, slow convergence and large dataset, etc., by using Dual Scale in DPTB to expand the receptive field, DSPE combines CNN with Transformer, which solves the above problems well. In addition, DP-ViT also maintains good detection accuracy, regression accuracy and robustness both in the case of adding noise and in the case of small sample datasets.

**5. Conclusions**

In this research, we propose a new DP-ViT network based on VIT, which can complete the target detection task for different types of sonar images. Referring to the Next-ViT structure, the Dual Scale Patch Embedding is innovatively introduced, which can effectively extract the local and global information from sonar images, enlarge the receptive field, and obtain richer multi-scale information. Dual Path Transformer Block combines the information of CNN and Transformer in a complementary way and introduces Conv-Attention to reduce the training parameters of the model, which improves the generalization of the model and reduces the required computing resources. In addition, Generalized Focal Loss is used to solve the problem of imbalance between positive and negative samples. The experimental results on the forward-look sonar dataset and the side-scan sonar image dataset show that DP-ViT has higher detection accuracy, faster detection speed, and anti-noise interference performance of the model, and solves the problems of many parameters,

slow convergence, and requires a large number of datasets caused by the Transformer structure. In the future, we will focus on reducing the parameters required for model training and deploy them on underwater vehicles to realize real-time sonar detection and improve the environmental awareness of AUV and USV.

**Author Contributions:** Methodology, Y.S.; Resources, H.Z.; Software, J.R. and H.X.; Visualization, C.X.; Writing—original draft, H.Z.; Writing—review & editing, Y.S. and G.Z. All authors have read and agreed to the published version of the manuscript.

**Funding:** This work was supported in part by the Science and Technology Project of Shaanxi Province Yinhan Jiwei Engineering Construction Co., Ltd. (SPS-D-15), in part by the Shaanxi Provincial Water Conservancy Science and Technology Program (2020slkj-5), in part by the Heilongjiang Provincial Natural Science Foundation key project (ZD2020E005) and in part by the Stable Supporting Fund of Acoustics Science and Technology Laboratory under Grant JCKYS2022604SSJS002.

**Data Availability Statement:** The data presented in this study are available on request from the Corresponding author.

**Acknowledgments:** Thanks to SenseCore for open source resources and projects on the Internet for deep learning, such as the mature mmdetection framework.

**Conflicts of Interest:** The authors declare no conflict of interest.

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
