# Peer review of "DP-ViT: A Dual-Path Vision Transformer for Real-Time Sonar Target Detection"

_remotesensing, doi:10.3390/rs14225807_

Round 1

Reviewer 1 Report

The paper deals exclusively with the instrumental problems obtained through the application of new algorithms in the processing of both front and side sonar signals.

It would be strongly advised that the results shown in the images be contextualized.

For exemple the bedforms in figure 2 are interesting, but there are not comments on the role of the new processing on these types of data.

Similar shortcomings in chapter 4.4, fig. 9 and fig. 10 where only the instrumental parameters are presented without highlighting the interpretative improvements.

Author Response

The reviewer gave more valuable comments on our manuscript, and we would like to take this opportunity to express our great appreciation on him/her as well as the comments. Followings are the changes made again in the new version together with our responses to the reviewer’s comments.

Reviewer 2 Report

The writing style needs to be improved. The quality of the figures is poor.

Author Response

(The authors gave the same response as above.)

Reviewer 3 Report

The article needs many corrections.

The "Introduction" part is not understandable. It does not introduce the problem. It only shows an outline of various problems. It needs to be rewritten.

How do you determine the exact position of an object in a sonar image? If so, what is this accuracy?

Which international standards were taken into account? IHO?

The authors separated the use of sonars.  “Forward look sonar can assist AUV to complete oil pipeline inspection, threat detection, mine hunting, and other tasks; Side scan sonar can help AUV find the location of wrecked ships and planes”

Why? On what basis?

The authors write about high-resolution images. What kind of resolution, at what level?

The authors write about “Traditional target detection methods”

What are the methods? Have these methods been compared with the proposed one?

The authors write: “AUV will  probably make wrong decisions..”

What decisions? This is incomprehensible.

In the introduction, the authors write about the 'transducer' and only in the next section do they explain what it is.

The authors write: “but the detection effect is much different from the accuracy of the method based on a deep neural network”

Why?

The authors write: “Many researchers have proposed many different sonar image target detection algorithms” What methods are these? Why have they not been quoted? For example, why dr David Polap was not quoted?

Generally, there is not enough related work.

The authors write: “for sonar image dataset with a small number of samples and low signal-to-noise ratio..”

What does this mean?

The authors write: “this paper makes experimental verification and comparative analysis on two datasets: forward look sonar and side scan sonar.”

It is as if the two sonars were being compared against each other. There are many such errors.

And suddenly a statement like this: “The images in the dataset are the original echo images of the forward look sonar after gain and coloring, but without polar coordinate transformation, which not only helps to keep more details of sound waves, but also facilitates the annotation and manual interpretation of the dataset.”

???

The experiment is incomprehensible. Perhaps some general methodological scheme would help?

How was confidence counted?

What can be seen in the drawings is not described. The tables are not described either.

Why were the same objects not used during the two sonar research?

The summary should include all the results and not just a statement that the method is useful.

Can the method be applied only to AUVs?

The results are not clearly described.

We can only see figures and tables but they are not described.

The article is chaotic and difficult to understand. It should be rewritten.

The work has many typos and grammatical errors. I am not an expert in English, but it seems to me that the article should be corrected in this respect.

Author Response

审稿人对我们的手稿提出了更有价值的意见,我们想借此机会对他/她以及这些意见表示我们的高度赞赏。以下是新版本中再次进行的更改以及我们对审稿人意见的回应。

Reviewer 4 Report

The manuscript presents a comparison of different DNN approaches to object classification of SSS and FLS images. The work is well written and the bibliography covers most aspects. However, the experimental part lacks basic information such as the FLS and SSS types, frequencies, distance to the objects, size of the objects, etc. AUV employed, geographical location (area) of the experiments, etc

Author Response

(The authors gave the same response as above.)

Round 2

Reviewer 1 Report

Paper improved in the text, remains unchanged in the representation of the reference side scan sonar images that could be clearer and accompanied by exhaustive captions

Author Response

(The authors gave the same response as above.)

Reviewer 2 Report

Writing style has been greatly improved, but figures are difficult to distinguish from each other due to their low resolution. Trust on the performance of the method is mostly based on the numerical results of comparisons with other methods.

Author Response

(The authors gave the same response as above.)

Reviewer 3 Report

All responses to previous comments should be included in the text - at least one sentence each. 

Author Response

(The authors gave the same response as above.)
